# Localized Cutaneous Nodular Amyloidosis in a Patient with Sjögren’s Syndrome

**DOI:** 10.3390/ijms24119409

**Published:** 2023-05-28

**Authors:** José María Llamas-Molina, Juan Pablo Velasco-Amador, Francisco Javier De La Torre-Gomar, Alejandro Carrero-Castaño, Ricardo Ruiz-Villaverde

**Affiliations:** 1Department of Dermatology, Hospital Universitario San Cecilio, Avda Conocimiento 33, 18016 Granada, Spain; josellamas94@gmail.com (J.M.L.-M.); pablo_r.m@hotmail.com (J.P.V.-A.); fjtogo@gmail.com (F.J.D.L.T.-G.); 2Department of Pathological Anatomy, Hospital Universitario San Cecilio, Avda Conocimiento 33, 18016 Granada, Spain; acarrero28@hotmail.com; 3Instituto Biosanitario de Granada, Ibs, 18014 Granada, Spain

**Keywords:** localized cutaneous nodular amyloidosis, plasma cells, Sjögren’s syndrome, autoimmune connective tissue disorders

## Abstract

Primary localized cutaneous nodular amyloidosis (PLCNA) is included in the primary forms of cutaneous amyloidosis along with macular and lichenoid amyloidosis. It is a rare disease attributed to plasma cell proliferation and deposition of immunoglobulin light chains in the skin. We present the case of a 75-year-old woman with a personal history of Sjogren’s syndrome (SjS), who consulted for asymptomatic yellowish, waxy nodules on the left leg. Dermoscopy of the lesions showed a smooth, structureless, yellowish surface with hemorrhagic areas and few telangiectatic vessels. Histopathology revealed an atrophic epidermis and deposits of amorphous eosinophilic material in the dermis with a positive Congo red stain. The diagnosis of nodular amyloidosis was made. Periodic reevaluation was indicated after the exclusion of systemic amyloidosis. PLCNA is often associated with autoimmune connective tissue diseases, and up to 25% of all PLCNA cases occur in patients with SjS. Therefore, in addition to ruling out systemic amyloidosis, screening for possible underlying SjS should be performed when the diagnosis of PLCNA is confirmed.

## 1. Introduction

The term ‘amyloidosis’ encompasses a range of conditions characterized by the extracellular deposition of amyloid, a fibrillar proteinaceous material. When amyloid deposition is generalized, it is referred to as systemic amyloidosis. In contrast, localized amyloidosis is limited to a specific location. Primary localized cutaneous amyloidosis (PLCA) can be classified into lichenoid, macular and nodular subtypes. Primary localized cutaneous nodular amyloidosis (PLCNA) is a rare disorder and is the least prevalent of the PLCA. In particular, PLCNA has a characteristic association with Sjögren’s syndrome (SjS). However, it is an uncommon cutaneous manifestation in this entity [1]. Herein, we present the case of a patient with primary SjS who subsequently developed PLCNA.

## 2. Case Report

A 75-year-old female with a 3-year history of multiple yellowish nodules on her left leg was referred to our dermatology department. These lesions were asymptomatic and had progressively increased in number and size. Her medical history included primary SjS, which had been diagnosed 6 years earlier. As a result, the patient had chronic xerostomia and xerophthalmia. The diagnosis of SjS was confirmed by elevated anti-SSA/Ro-52 titers, a pathologic Schirmer’s test and severe submaxillary gland dysfunction on salivary gland scintigraphy.

Physical examination revealed waxy, yellowish and raised nodules (*n* = 4) of various sizes (the largest being 4.0 cm in diameter) located in the left pretibial region (Figure 1A). The consistency was firm, and the borders were well-defined (Figure 1B). No other similar lesions were found at any other site. Dermoscopy showed a smooth surface with no obvious structures, yellowish–brown color and irregular borders. There were hemorrhagic areas mainly in the periphery and few vessels with telangiectatic appearance (Figure 2). Histopathology revealed an atrophic epidermis and deposits of amorphous eosinophilic material in the superficial and deep dermis as well as in subcutaneous cellular tissue, vessel walls and eccrine glands. Clusters of plasma cells were also visible in some areas of the dermis and subcutaneous cellular tissue. Eosinophilic material stained positive for Congo red and exhibited characteristic apple-green birefringence under polarized light (Figure 3). Positive results for immunoglobulin κ and λ light chains were obtained using immunohistochemical techniques. On this basis, a diagnosis of nodular amyloidosis was established.

Laboratory tests showed ANA positivity (>1/160) with a fine speckled pattern and elevated anti-SSA/Ro-52 (≥240 U/mL). The blood cell count test and proteinogram were normal, and no M protein was detected in the serum nor were Bence Jones proteins detected in the urine. No amyloid deposition was seen in the abdominal fat punch biopsy. The bone marrow biopsy was also normal, with negative Congo red staining and a plasma cell population of approximately 5% with no evidence of monotypic expression. Therefore, systemic amyloidosis was ruled out, and since the AL amyloid deposits were exclusively confined to the skin, the diagnosis of PLCNA was made. Therapeutic options were offered to the patient but she declined them because the lesions were asymptomatic. Thus, it was decided not to initiate any treatment for the time being. It was agreed to reevaluate the situation periodically. At the six-month follow-up, the previous lesions were ulcerated with mild to moderate pain and a new lesion had appeared on the posterior aspect (Figure 4). Therefore, the patient is undergoing regular local dressings of her lesions. It has been agreed with her to start treatment with UVA1 phototherapy once the ulceration has healed. She continues to show no evidence of systemic amyloidosis.

## 3. Discussion

The amyloid substance is a fibrillar proteinaceous material composed of two components: a common one derived from the serum amyloid component and a specific one that defines the type of amyloidosis [2]. Specifically, the amyloid substance present in PLCNA is of the AL type, as occurs in primary systemic amyloidosis. In these entities, the amyloid is derived from light chains produced by plasma cells. In contrast, in the other two forms of PLCA, known as macular amyloidosis (MA) and lichenoid amyloidosis (LA), the amyloid substance is of the K type. This observation is due to the fact that in MA and LA the amyloid is derived directly from keratinocytes. Thus, in these two localized forms, the amyloid deposition is located in the upper dermis. In PLCNA, which is the least common form of PLCA, the amyloid affects the dermis, subcutaneous tissue and blood vessels and is usually accompanied by plasma cell infiltrates [1].

PLCNA presents clinically as firm, indurated, waxy plaques or nodules of yellowish, brownish or violaceous color. It is not uncommon for the lesions to have a purpuric appearance, and sometimes, telangiectasias are seen on the surface. Atypical presentations simulating a lymphatic malformation [3], milia [4] or a bullous disease [5] have also been reported. PLCNA lesions may be single or multiple and most commonly occur in the acral region, head, neck and extremities [6]. Occasionally, cases of PLCNA have been described in atypical locations, such as the scalp [7], tongue [8], areola [9] or toe [10]. Although plaques and nodules are usually asymptomatic, patients occasionally report tenderness, pruritus or even pain. This symptomatology may be related to the mass effect and compression of adjacent tissues. Their tendency to ulcerate may also be explained by this fact [11]. Entities that clinically present with indurated nodules or plaques of yellowish, orange or pink color are included in the differential diagnosis of PLCNA. When found on the extremities, trunk, face or acral areas, which are the most common sites, the differential diagnosis should include cutaneous B and T lymphomas, cutaneous lymphocytoma cutis, necrobiosis lipoidica, sarcoidosis, granuloma annulare or pretibial myxedema, among others. However, because it can occur in many different locations and its clinical presentation can be atypical, the differential diagnosis may include an even broader group of conditions [12].

There is a paucity of literature on PLCNA dermoscopy. As in our case, the most common pattern appears to be a yellowish background without structure, interrupted by whitish scar-like lines with some hemorrhagic areas and vessels resembling fine telangiectasias. Several authors have postulated that the yellowish–orange color seen in PLCNA lesions may correspond to nodular aggregates of amyloid in the reticular dermis and subcutaneous tissue. Such aggregates may sometimes involve the adnexa and vessel walls. It may appear as a diffuse yellowish background or as rounded yellowish or orange structures. Therefore, PLCNA should be considered in the differential diagnosis of yellow–orange lesions on dermoscopy. This group includes several histiocytic and granulomatous diseases, both inflammatory, such as xanthogranuloma or sarcoidosis, and infectious, such as leishmaniasis or lupus vulgaris. In PLCNA, the unstructured yellow aggregates differ from the small yellow dots of these granulomatous diseases, which have a more pronounced inflammatory infiltrate [13]. When the PLCNA lesion is on the face, multiple, tiny, uniformly sized white dots are usually seen. As with other facial lesions, this may be explained by the presence of facial eccrine gland orifices [14]. As for the bright white lines, they may be related to a change in the composition or orientation of the collagen. They can be seen more clearly with polarized light [15].

Histopathology is considered the gold standard for the diagnosis of PLCNA. A nodular deposition of hyaline and eosinophilic material is usually seen throughout the thickness of the dermis. Such material may also be found in small vessel walls, adnexa and subcutaneous tissue. Variable plasma cell infiltration may occur within amyloid deposits adjacent to the vasculature. Eosinophilic material corresponding to amyloid substance is stained with Congo red and shows a characteristic brick-red deposition. Under polarized light microscopy, an apple-green birefringence is seen. The subtype of amyloid deposits must then be determined, which reveals the κ and/or λ light chain [13]. Planas-Ciudad et al. [16] reported a case of PLCNA with dystrophic calcifications on ultrasound study and with subsequent histologic confirmation. Dystrophic calcifications are rarely seen in PLCNA but have been described in cases of localized amyloidosis of other organs. In cases of PLCNA with extensive or deep involvement, soft tissue imaging, preferably magnetic resonance imaging (MRI), should be obtained [11].

To establish the diagnosis of PLCNA, primary systemic amyloidosis must first be excluded. This is because the histopathologic findings of PLCNA are virtually identical to those seen in the associated skin lesions of the primary systemic form, in which the amyloid substance is also of AL type. A thorough physical examination and a blood test, including a complete blood count and metabolic panel, are recommended. The presence of a monoclonal plasma cell population outside the skin lesions is assessed by electrophoresis and immunofixation in serum and urine. Thus, the absence of the monoclonal protein (M) in the serum and the absence of Bence Jones proteinuria are indicative of PLCNA. Likewise, monoclonal plasma cells should not be present in the bone marrow examination. Some authors also suggest initial imaging studies to assess systemic involvement, such as positron emission tomography (PET). An electrocardiogram may be ordered to exclude cardiac involvement [11]. In addition, patients with PLCNA should be periodically reevaluated because there is some risk of developing systemic amyloidosis. Initially, the risk of progression to systemic amyloidosis in patients with PLCNA was thought to be close to 50%. More recently, however, based on a series of a few patients followed over time, it has been estimated to be about 1–7%, although the rate of paraproteinemia may be as high as 40% [17].

The management of PLCNA is particularly challenging given the relative infrequency of diagnosis and lack of prospective studies. There are no clear management recommendations and no treatment of choice. In fact, the evidence on this topic comes from case reports. The treatments used are often ineffective and local recurrences may occur. The most commonly reported therapies for PLCNA are aimed at removing or improving the appearance of the lesion. Among these, surgical excision is one of the most commonly used treatments. Previous therapies have included surgical excision, dermabrasion, electro-desiccation and curettage. The results of these therapies are variable, and there is a high rate of lesion recurrence. In addition, it has not been effectively treated with other physical interventions, such as cryotherapy [18]. There have been good responses to CO_2_ laser treatment in the PLCNA, with tissue fragility during treatment and subsequent atrophy being the reported complications [19,20,21]. There are also two published reports showing excellent responses to combined surgery and CO2 laser [22,23]. Pulsed dye laser (PDL) has been shown to be beneficial in the treatment of PLCNA.

Systemic treatment should be considered in extensive or multifocal cases or when local modalities fail to control the disease [11]. However, the choice of appropriate immunosuppressive treatment appears to be complex, as PLCNA tends to respond poorly to immunosuppressive therapies. This may be because systemic immunosuppressive agents are ineffective against local secretion by cutaneous plasma cells [24]. Tong et al. reported a case of a patient who failed multiple immunosuppressive therapies and finally responded to cyclophosphamide. The use of cyclophosphamide in PLCNA is supported by its efficacy in myeloma and systemic AL amyloidosis [25]. Similarly, Khan et al. presented a case of a good response to the combination of bortezomib and dexamethasone in a 64-year-old woman. The patient had recurrent PLCNA and had previously undergone several local therapies with no response. The validity of this therapy was based on the good results obtained with the combination of proteasome inhibitors and corticosteroids in the treatment of systemic immunoglobulin light chain amyloidosis (AL) [11]. Intralesional methotrexate may be an option for patients with localized lesions who are poor candidates for surgery or invasive procedures [26]. Further studies are needed to determine the role of systemic therapy in PLCNA [18].

SjS is a chronic autoimmune lymphoproliferative disease characterized by inflammation and dysfunction of the secretory glands, primarily the salivary and lacrimal glands, resulting in dry mouth and eyes. SjS may be primary or associated with another autoimmune connective tissue disorder and usually begins to manifest in the fourth and fifth decades of life. Its prevalence is nearly 60 cases per 100,000 individuals, with a ratio of 9:1 (F:M) [27]. Exocrine glands are infiltrated by T and B cells and progressively destroyed by the cellular and humoral response. The epithelial cells of the glands are currently thought to play an important role in the pathogenesis of SjS as they are the triggers of immune activation. Thus, epithelial cells participate in this activation by expressing ribonucleoprotein complexes (i.e., Ro/SSA and La/SSB), interacting with T cells, and producing cytokines. Although T lymphocytes play an important role in pathogenesis, B lymphocytes are the main cells involved and are also implicated in SjS-associated non-Hodgkin’s B-cell lymphoma. There are increased levels of cytokines normally associated with B lymphocytes, such as IL-6 and IL-10. In addition, serum B-cell activating factor (BAFF) levels are elevated, and germinal center-like structures have been identified in the salivary glands of patients with SjS. Ongoing B-cell hyperactivity and light chain-producing plasma cells can lead to amyloid deposits in the skin [28]. This would explain the development of PLCNA in patients with SjS. Although PLCNA is not a common cutaneous manifestation in patients with SjS, nearly 25% of reported cases have been associated with this entity [29]. There is no underlying hematologic dyscrasia in PLCNA, and it is currently considered a reactive rather than a neoplastic disorder [2]. However, occasional plasma cell monoclonality has been described [30,31]. This fact justifies that PLCNA may be part of the spectrum of lymphoproliferative disorders that occur in SjS. On the other hand, because plasma cells are found only in the skin and not in the bone marrow, some authors consider PLCNA to be an extramedullary plasmacytoma in which plasma cells locally produce light chains, leading to fibril formation and subsequent amyloid deposition [31].

At present, it is not known in detail whether there is any peculiarity in the clinical presentation of PLCNA in patients with SjS. In this regard, the article by Yoshida et al. [32] is noteworthy, in which a comparison was made between Japanese patients with PLCNA with or without associated SjS. They observed that in patients without SjS, head involvement was much more common than the trunk or lower extremity involvement, which seems to be more common in patients with PLCNA and SjS. The mean age of patients in both groups was approximately 60 years. Gender varied considerably, with a clear female predominance in the group with associated SjS. In patients without SjS, however, there were almost twice as many men as women [32]. These gender differences may be explained by the fact that SjS is much more common in women [1]. Previous series of patients with PLCNA have shown no gender differences [18] or even a male predominance [33]. These series included cases with and without comorbidities. Both series are retrospective with long-term patient follow-up. The study of Woollons et al. [18] includes 15 cases. However, it is not stated whether any of them also suffered from SjS. The series by Moon et al. [33] includes 16 patients, two of whom also had SjS but it is not specified which of them had SjS. Therefore, it is likely that when PLCNA and SjS coexist, the age and gender will not vary, thus occurring in female patients in the seventh decade of life. The location of the lesions would be different as it is very common for patients with SjS to have lesions on the trunk and/or lower extremities rather than on the acral parts or face.

The majority of case reports documenting PLCNA in association with autoimmunity have been in the setting of SjS. However, PLCNA has also been described in association with other connective tissue diseases, such as CREST (calcinosis, Raynaud’s syndrome, esophageal involvement, sclerodactyly, telangiectasia) [34,35], systemic sclerosis [14,36] and systemic lupus erythematosus [7]. This suggests that plasma cell dysregulation in the skin may be related to the autoimmune imbalance inherent in connective tissue diseases. Limited systemic sclerosis (including CREST syndrome) appears to be the entity most commonly associated with PLCNA after SjS. As with the latter, PLCNA in limited systemic sclerosis is more common in postmenopausal women and in the lower extremities. Atzori et al. suggested that the progressive hardening of soft tissues, loss of appendages and macro- and micro-vascular involvement that occurs in scleroderma, particularly in the distal extremities, may contribute to the isolation of a pool of monoclonal plasma cells in the dermis. The light chains released by these cells would accumulate and not be cleared due to poor circulation [13].

## 4. Conclusions

PLCNA is a rare disease that is closely related to SjS. Therefore, in addition to ruling out systemic amyloidosis, a confirmed diagnosis of PLCNA should prompt the clinician to look for associated SjS in addition to other connective tissue diseases. Management of these patients should include regular screening for systemic amyloidosis, although the risk of progression to systemic amyloidosis is low. Treatment is complex and often ineffective, and local recurrences are not uncommon.

## Figures and Tables

**Figure 1 ijms-24-09409-f001:**
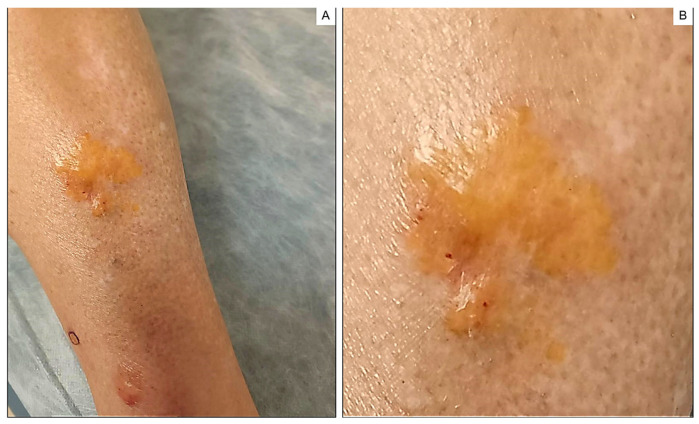
(**A**) Waxy, yellowish, raised nodules with well-demarcated margins on the anterior pretibial aspect of the left leg. (**B**) Close-up of larger lesion at higher magnification.

**Figure 2 ijms-24-09409-f002:**
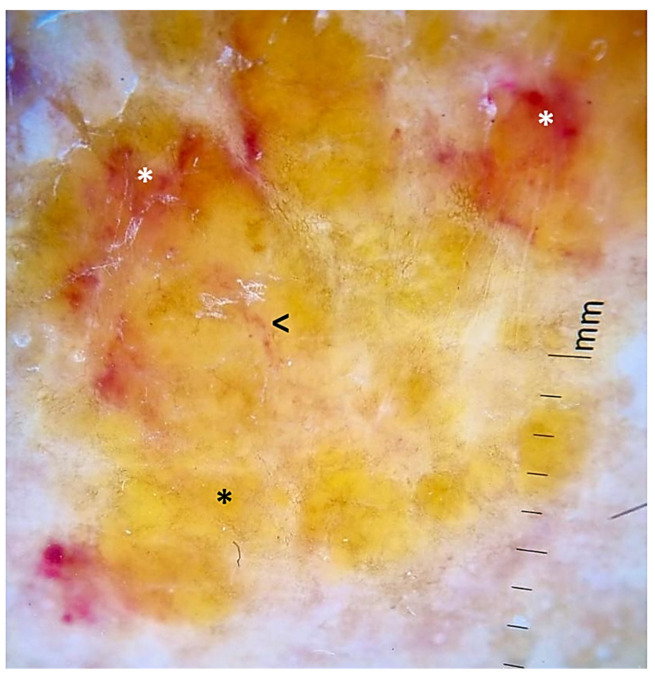
Dermoscopy (DermLite DL4©). Lesion with yellowish and smooth surface, without apparent structures (black asterisk). Some telangiectatic vessels (arrowhead) and areas of hemorrhagic appearance in the periphery (white asterisk) were also observed.

**Figure 3 ijms-24-09409-f003:**
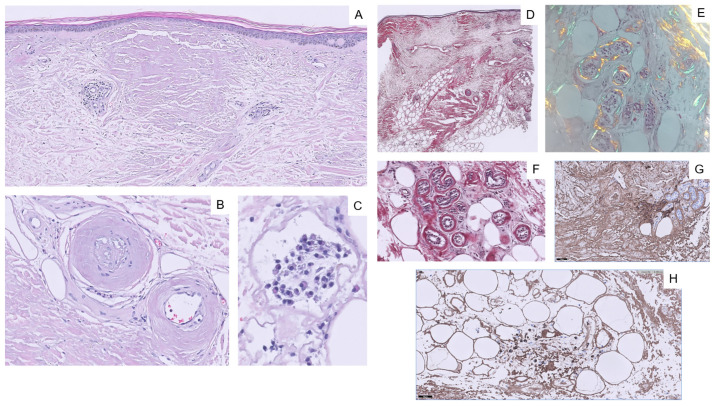
Histopathology. (**A**) H&E, ×10. Atrophic epidermis and deposits of amorphous eosinophilic material in the superficial and deep dermis. It is separated from the epidermis by a thin line of collagen. (**B**) H&E, ×20. Deposit of amorphous eosinophilic material affecting the vascular walls. (**C**) H&E, ×40. Cluster of plasma cells. (**D**,**E**) Eosinophilic material stains with Congo red positive stain and acquires the characteristic apple-green birefringence under polarized light. (**F**) Congo red stain around the eccrine glands. (**G**) Immunohistochemical techniques for λ light chain. (**H**) Immunohistochemical techniques for κ chain.

**Figure 4 ijms-24-09409-f004:**
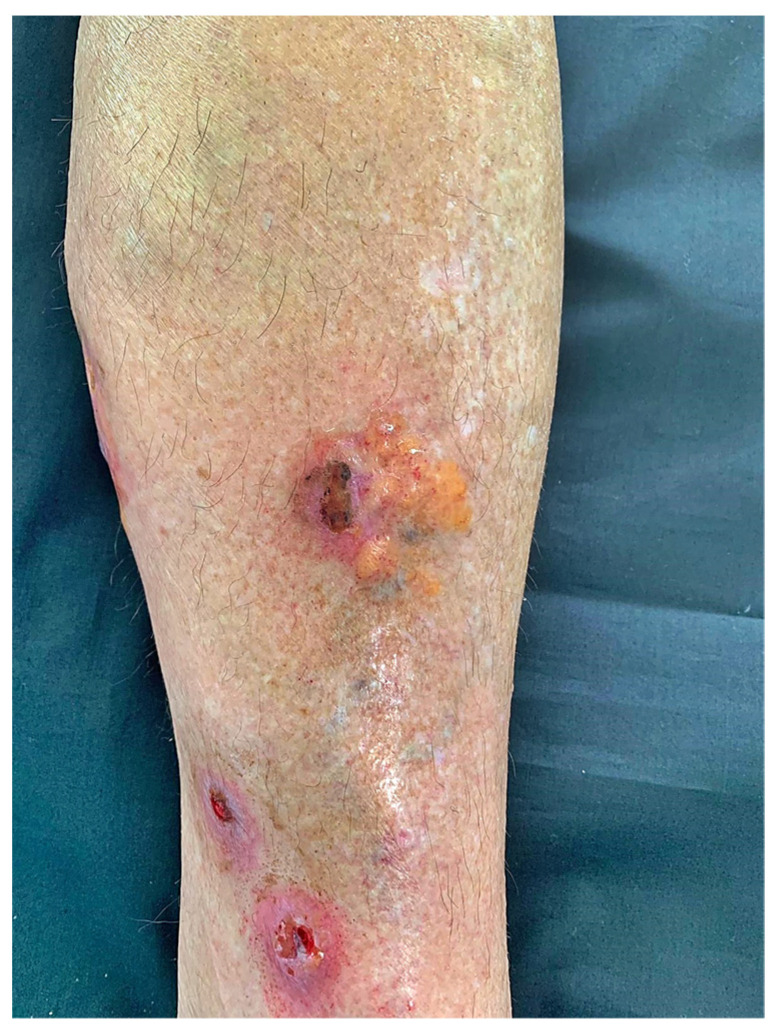
Lesions became ulcerative at annual check-up.

## Data Availability

Data are available upon reasonable request to authors.

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
