# Peer review of "Localized Cutaneous Nodular Amyloidosis in a Patient with Sjögren’s Syndrome"

_ijms, 2023, doi:10.3390/ijms24119409_

Round 1

Reviewer 1 Report

Very interesting article but in my opinion discussion is too long and  should be shortened, the section describing SJS is redundant.

Author Response

The authors would like to thank you for your feedback as a reviewer of our article.
Regarding the shortening of the discussion, we would like to point out that the discussion is of this length in order to meet the minimum word count set by the editor. We also feel that this is a topic that can be explored in depth, so we do not feel that this section should be shortened. The relationship between the PLCNA and the SJS is special. We believe that this is what we should focus most on in our discussion and what may be of most interest to readers. Therefore, we believe that all of the information presented about SjS should be considered without eliminating any part of it.

The authors have worked hard on the article and are pleased that the reviewer found it interesting. We hope that you will consider our opinion and the reasons why the discussion should not be shortened.

We look forward to hearing from you. 
Yours sincerely
The authors

Reviewer 2 Report

A well-written relevant case-report for a rare condition. 

Author Response

We, the authors, would like to thank you very much for your kind words regarding our article. We have worked hard on the paper and we are pleased that the reviewer likes it.

Reviewer 3 Report

This is a exceptionally well researched and presented case report of a rare dermatologic condition.  The disease process is well documented with photographs, dermoscopy, histopathology, immunohistochemical evaluation and description of pathophysiology and patient history. 

Author Response

We, the authors, are very grateful for your kind words towards our article. We have worked hard on this paper, so we are pleased that the reviewer likes it.

Reviewer 4 Report

The case report is clearly presented. Excellent discussion of the article. The submitted article is useful in clinical practice, too.

Author Response

We, the authors, are very grateful to you for your kind words in response to our article. We are delighted that the reviewer likes the paper, as we have worked hard on it.